# Do Obese Children Achieve Maximal Heart Rate during Treadmill Running?

**DOI:** 10.3390/sports7010026

**Published:** 2019-01-19

**Authors:** Sveinung Berntsen, Elisabeth Edvardsen, Shlomi Gerbi, Magnhild L. Kolsgaard, Sigmund A. Anderssen

**Affiliations:** 1Department of Public Health, Sport and Nutrition, Faculty of Health and Sport Sciences, University of Agder, P.O. Box 422, NO-4604 Kristiansand, Norway; gerbishlomi@gmail.com; 2Department of Sports Medicine, Norwegian School of Sport Sciences, 0806 Oslo, Norway; elisabeth.edvardsen@nih.no (E.E.); s.a.anderssen@nih.no (S.A.A.); 3Department of Pulmonary Medicine, Oslo University Hospital, 0424 Oslo, Norway; 4Department of Paediatrics, Oslo University Hospital, 0424 Oslo, Norway; UXPOMA@ous-hf.no

**Keywords:** play, exercise, fitness, physical activity, exercise testing

## Abstract

Objective: Maximal heart rate (HR) is commonly defined as the highest HR obtained during a progressive exercise test to exhaustion. Maximal HR is considered one of the criteria to assess maximum exertion in exercise tests, and is broadly used when prescribing exercise intensity. The aim of the present study was to compare peak HR measurements during maximal treadmill running and active play in obese children and adolescents. Design: Comparison of peak heart rate during active play vs. maximal treadmill running in 39 (7–17 years old, 18 males) obese children and adolescents. Methods: Heart rate was recorded during intensive active play sessions, as well as during a progressive running test on a treadmill until exhaustion. HR, respiratory exchange ratio (RER), and oxygen uptake were continuously measured during the test. The criteria for having reached maximal effort was a subjective assessment by the technician that the participants had reached his or her maximal effort, and a RER above 1.00 or reporting perceived exertion (RPE) above 17 using the Borg-RPE_6–20_-Scale. Results: Thirty-four children had a RER ≥1.00, and 37 reported a RPE ≥ 17. Thirty-two children fulfilled both criteria. During active play, peak HR was significantly (*p* < 0.0001) increased (4%) (mean and 95% confidence intervals; 204 (201, 207) beats/min), compared to during maximal treadmill running (196 (194, 199) beats/min), respectively. Conclusion: The results of the present study indicate that peak heart rate measurements during progressive running to exhaustion in obese children and adolescents cannot necessarily be determined as maximal heart rate.

## 1. Introduction

Behavioral lifestyle interventions focusing on physical activity may have the potential to reduce level of overweight and obesity in children and adolescents [1]. Encouraging or participating in active play sessions has been reported to stimulate to increased energy expenditure [2] or physical activity level among obese children [3]. In order to encourage lifestyle changes in children, physical activity or exercise training should be enjoyable, age-specific, and adapted to the child’s fitness level [4]. To tailor exercise intensity to the child’s cardiorespiratory fitness level or to prescribe an intensity zone based on maximal heart rate (HRmax), e.g., above 80% of their HRmax, cardiopulmonary exercise testing with measures of peak HR is common and recommended [5,6]. HRmax is commonly defined as the highest HR obtained during a progressive exercise test to exhaustion [7]. Equations for predicting HRmax have been developed [8,9] and validated in children and adolescents performing maximal treadmill running or bicycling on a cycle ergometer [10,11,12,13]. Though, unmotivated performances during maximal testing may influence children’s test results [13,14,15], and peak HR attained during, e.g., treadmill running, may not reflect true maximal HR. When underestimating HRmax, exercise intensity during more enjoyable activities may be performed at lower exercise intensity than actually supposed. Comparisons of peak HR during active play and more traditional exercises, like treadmill running, have not been carried out. The *aim* of the present study was, therefore, to compare peak HR measurements during maximal treadmill running and active play in obese children and adolescents.

## 2. Methods

Obese and overweight children and adolescents (hereafter called children) from Oslo were referred by their physician or school nurse to the outpatient pediatric clinic for participation in a multidisciplinary obesity management program at the Department of Pediatrics at Oslo University Hospital in Norway [16]. The children were examined by a pediatrician and included if they were obese according to the Norwegian percentile diagram (body mass above the 97.5^th^ percentile for height) [17]. Children 7–17 years old, without an overt or organic disease (in which there were anatomical or path physiological changes in some bodily tissue or organ) causing the obesity, medical conditions that could limit the ability to be physically active, and receiving medication which could affect growth or weight control, were included. A detailed description of the inclusion procedures, methods, and intervention are presented elsewhere [3,16]. 

Of the 120 referred children, 60 subjects participated in physical activity intervention [3], and 39 children, with both treadmill test and heart rate measurements during active play sessions, attended the present study. The children included in the present study were representative of 7–17-year-old subjects in “The Oslo Adiposity Intervention Study” with respect to gender, age, pubertal status and body mass above the 97.5^th^ percentile for height.

The study was approved by the Regional Committee for Medical and Health Research Ethics, South East, and the Data Inspectorate of Norway (S-04313). Written informed consent was obtained from all children and their respective parents.

Body mass was measured wearing light underwear to the nearest 0.1 kg (Seca 770, Hamburg, Germany). Height was measured by a stadiometer to the nearest 0.5 cm. Body composition was measured by an experienced technician by dual-energy X-ray absorptiometry (DXA; GE-Lunar Prodigy, Madison, WI, USA). Participants were scanned from head to toe in a supine position, providing fat mass for the total body, as well as separately for arms, legs, and trunk. Test–retest analyses from 30 scans in 15 children and further details have been reported previously [3]. Physical activity was objectively recorded by the ActiGraph 7164 accelerometer (LLC, Fort Walton Beach, FL, USA) for seven consecutive days before start of the intervention. The output was sampled every 20 s and presented as mean counts per minute (cpm). Sequences of ≥10 min with consecutive zero counts were automatically deleted. As in other studies, moderate-to-vigorous intensity physical activity (MVPA) was defined as all physical activity above 2000 cpm [18,19].

A progressive maximal treadmill test (Woodway, WI, USA) was performed after five minutes familiarization while walking on the treadmill. The maximal test started at four kilometers per hour (km∙h^−1^) with an inclination of 0%, increasing the work load (individualized increase in speed, 0.5 km∙h^−1^ per minute, and/or inclination, 1% per minute) until exhaustion. During the last part of the test, the participant’s effort was largely encouraged by the technician until voluntary termination. Chest-measured HR was recorded continuously during the test (Polar Sports Tester 3000, Polar Electro KY, Kempele, Finland), with the highest recorded HR defined as peak HR. Minute ventilation (${\overset{\cdot }{V}}_{E}$), respiratory exchange ratio (RER), and ${\overset{\cdot }{V}O}_{2}$ were continuously measured using the Sensor Medics, Vmax Spectra (Yorba Linda, CA, USA). The rating of perceived exertion (RPE) was obtained using the Borg-RPE-Scale_6–20_ [20]. The criteria for reached maximal effort was a subjective evaluation by the technician that the child had reached his or her maximal effort, and a RER above 1.00 or RPE above 17.

The five-month exercise intervention consisted of 60 min guided active play, twice a week, with different activities focusing on coordination, flexibility skills, and self-esteem in the first four weeks. Thereafter, team play, endurance-, and strength-type activities, such as body weight calisthenics, ball games, wrestling, or fun-related movements were included. Duration and intensity of each activity varied. The mean attendance in the sessions was 60%. During the active play sessions, chest-measured HR was recorded using Polar Vantage (Polar Electro KY, Kempele, Finland). The highest recorded HR (recorded within 30 s intervals) during any of the active play sessions (60 min each) was defined as peak HR during active play [3]. 

Demographic data are given as mean with standard deviation (SD), unless otherwise stated, and results as mean with SD or 95% confidence intervals (CI). Independent *t*-test was used to analyze differences between groups. Paired sample *t*-test was used to analyze differences between peak HR measurements during active play sessions and maximal treadmill running. Statistical significance level was set to 5%. Statistical analyses were performed with Statistical Package for Social Sciences Version 21.0 (SPSS, Chicago, IL, USA).

## 3. Results

Physical characteristics of the participants are shown in Table 1. Girls were significantly lower (*p* = 0.01) and had significantly higher percentage body fat (*p* = 0.03). Boys participated in a significantly higher amount of MVPA (*p* < 0.01). 

Data from the maximal treadmill test are presented in Table 2. The RER at exhaustion was 1.03 (SD, 0.08), and was significantly higher (*p* = 0.04) among girls. Thirty-four out of 39 children had a RER above 1.00. Self-reported perceived exertion after running using the Borg-RPE-Scale was 18 (2) on average, with 37 children reporting above 17. Thirty-two children reported both above 17 on the Borg-RPE-Scale and had a RER above 1.00. The test duration on the treadmill was 8 (2) min with a final speed and inclination at exhaustion of 7.3 (0.6) km∙t^−1^ and 7 (2)%, respectively.

Peak HR was 4% higher and significantly (*p* < 0.0001) increased during active play (mean and 95% confidence intervals; 204 (201, 207) beats per min (bpm)) compared to during maximal treadmill running (196 (194, 199) bpm), respectively (Figure 1).

## 4. Discussion

In the present study, peak HR increased during active play compared to during maximal treadmill running in obese children. To our knowledge, this is the first study to compare obtained peak HR during different test conditions, like laboratory-based and play-based (in the field). 

Despite that exercise testing using a treadmill or stationary bike in the laboratory is common when assessing physiological variables, there is no consensus in relation to protocols for assessment of maximal effort or HR in children [1]. Together with age, gender, type of exercise, state of health, fitness, and environmental conditions [13], the type of exercise protocol could affect peak HR. The average peak HR from active play sessions of 204 bpm and 194 bpm during treadmill testing cannot necessarily be compared with previous studies reporting peak HR of 193 bpm [21] to 202 bpm [22] during treadmill running.

An exercise test with the purpose of measuring a physiological peak level is dependent on the participant to provide the maximum of what is physiologically possible for them. Lack of motivation could be a factor that makes the participants terminate the test before reaching their physiological peak level [15]. For instance, testing in a laboratory can be more competitive and unpleasant compared to other forms of testing, and fitness testing can be regarded as “demanding, embarrassing and highly uncomfortable” and could develop a negative attitude toward physical activity [2]. This is in deep contrast to free play, where the aim is to develop a positive attitude to activity and the perspective of fun, where the motivation come from within the player [23]. Also, motor skills efficiency and a protocol that can match the motor skills are important determinants of peak HR [12]. A peak HR that was 4% higher during active play may indicate that the laboratory-based protocol in the present study failed in motivating children in a similar way to the active play sessions, explaining the higher peak HR. This corresponds to a previous study arguing that a high level of perceived competence enables individuals to endure the discomforts of the maximum effort and to achieve higher performance [15].

Until now, in several studies [21,24,25,26], measures of children’s peak HR using treadmill and cycle ergometer [13] have been carried out, but hardly any literature exists showing peak HR in relation to children in active play or out of the laboratory. With the present study showing that obese children can reach significantly higher peak HR during active play, compared with treadmill running, the role of conventional laboratory testing to target exercise intensity zones in children could be questioned. Protocols adapted to children’s motor skills and motivation should be developed as peak HR is a commonly used variable in clinical medicine and physiology prescribing exercise intensity in rehabilitation and disease prevention programs [9]. The implications of underestimating peak HR could be to prescribe lower exercise intensity than initially planned. The American College of Sports Medicine (ASCM) recommends a moderate and vigorous exercise intensity level for children [7]. For instance, if the treadmill protocol would be the foundation for establishment of exercise intensity, a recommendation to exercise at moderate intensity (using ASCM intensity scale [27]) would be to exercise with a HR between 125 bpm and 147 bpm for the average child in the present study. The fact that the active play protocol in the present study showed that the same average child had a significantly higher peak HR, illustrates that the average child with a HR of 135 bpm during exercise may be classified as low intensity, instead of moderate intensity [28]. A consequence of exercising at lower intensity may result in reduced improvements in fitness. The 4% difference between active play and treadmill running may change the distribution of work in both rehabilitation and disease prevention programs, as it is important, in an exercise program, to determine and report the correct amounts of volume, frequency, and intensity [29]. 

The main strengths of the present study are the experienced test leaders, a relatively large number of participants, and several play-based exercise sessions. Whether use of an alternative treadmill protocol may have resulted in improved peak HR values compared to the present protocol cannot be excluded. Warming up for at least 20 minutes followed by two to three uphill running sessions may give higher peak HR values compared to the present treadmill protocol [30].

## 5. Conclusions

In conclusion, obese children tested for peak HR in two different environments had significantly higher peak HR during free play compared to treadmill running. The results of the present study indicate that peak HR measurements during progressive running to exhaustion in obese children cannot necessarily be determined as maximal HR.

## 6. Practical Implications

How to report and interpret exercise intensity seems important in training programs for obese children and adolescents.The type of exercise as well as setting may influence exercise intensity.Active play may result in higher exercise intensity compared to treadmill running in obese children and adolescents.

## Figures and Tables

**Figure 1 sports-07-00026-f001:**
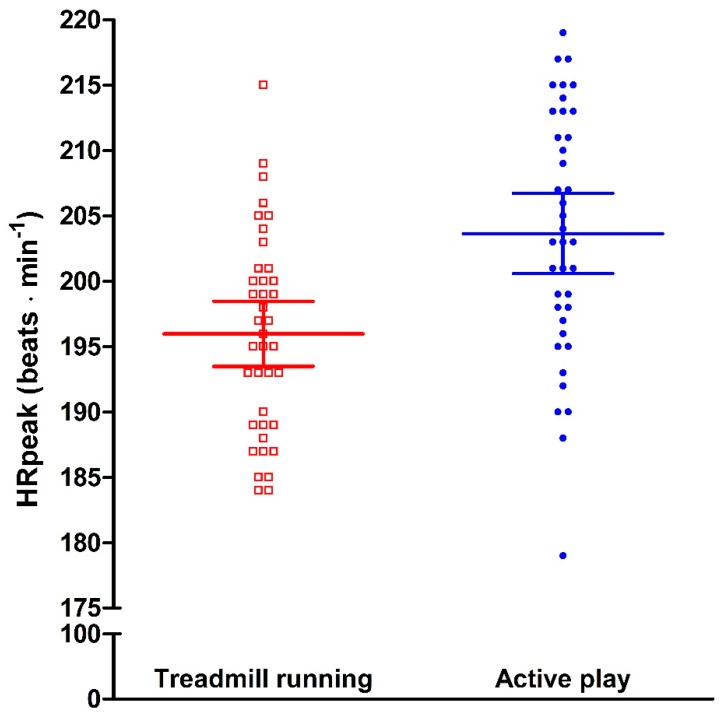
HR peak during both treadmill running and active play. Individual values with corresponding mean and 95% confidence intervals.

**Table 1 sports-07-00026-t001:** Baseline physical characteristics of the 39 participating children presented by gender. Data are given as mean and standard deviation in parentheses.

	Girls (*n* = 21)	Boys (*n* = 18)	*P*-value **
Age (years)	12 (2)	13 (2)	0.10
Body mass (kg)	70 (19)	75 (17)	0.15
Height (cm)	154 (10)	161 (10)	0.01
Percentage of body fat (%)	49 (5)	45 (6)	0.03
MVPA (min·day^−1^)	43 (14)	81 (22)	<0.01

Abbreviations: moderate-to-vigorous intensity physical activity (MVPA); ** *P*-values for any differences between groups.

**Table 2 sports-07-00026-t002:** Physiological responses during maximal treadmill running presented by gender. Data are given as mean and standard deviation in parentheses.

	Girls (*n* = 21)	Boys (*n* = 18)	*P*-value **
Peak $\overset{\cdot }{V}E$ (1∙min^−1^)	82.4 (25.0)	87.0 (20.1)	0.55
Peak RER	1.06 (0.07)	1.00 (0.09)	0.04
Peak RPE	18 (2)	18 (1)	0.74
Peak HR (beats∙min^−1^)	195 (8)	197 (7)	0.58
Speed at termination (km∙t^−1^)	7.4 (0.6)	7.2 (0.6)	0.23
Inclination at termination (%)	7.1 (1.6)	6.7 (2.1)	0.51
${\overset{\cdot }{V}O}_{2}$ peak (ml∙kg^−1^∙min^−1^)	33.5 (5.1)	34.7 (7.1)	0.53

Abbreviations: $\overset{\cdot }{V}E$, minute ventilation; RER, respiratory exchange ratio; RPE, rating perceived exertion; HR, heart rate; ${\overset{\cdot }{V}O}_{2}$, oxygen uptake. ** *P*-values for any differences between groups.

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
