# Peer review of "Do Obese Children Achieve Maximal Heart Rate during Treadmill Running?"

_sports, 2019, doi:10.3390/sports7010026_

Round 1
Reviewer 1 Report
I think this is a very good paper and have minimum comment. There is a typo on Figure 1 (Activ ->Active) and I would suggest having a strength and limitation paragraph in the discussion
Author Response
Reviewer 1
I think this is a very good paper and have minimum comment. There is a typo on Figure 1 (Activ ->Active) and I would suggest having a strength and limitation paragraph in the discussion
Thank you, we have corrected Activ to Active and added a strength and limitation paragraph in the revised manuscript.
Reviewer 2 Report
The purpose of the study was to compare peak HR measurements during maximal treadmill running and active play in obese children and adolescents. The study is nice and well written. However, I have a major concern mainly with the methodology.
Major comments
Could the authors give more details about the methodology, for example a description of the progressive treadmill test (speed, and inclination increases, step duration).
Did you test the reproducibility of the protocols?
Did the authors realize a familiarization session? a warm-up?
Thank you for giving some details about all that.
Minor comments
L60, P2: “ from children 7-17 years…” the sentence is not clear. Please to rewrite.
L83: “cpm” to explain
L93: The objective criteria for maximal effort included at least 2 of the following: (1) increased workload without corresponding increase in oxygen consumption; (2) respiratory exchange quotient”. What about the plateau of VO2? Do you observed a plateau of VO2 at the end of the treadmill test in your children?
L94,P3: How many sessions of 60-min the children realized?
L150,P3: write “where the motivation” instead of “where to”
Author Response
Reviewer 2
Major comments
Could the authors give more details about the methodology, for example a description of the progressive treadmill test (speed, and inclination increases, step duration).
More details with respect to the treadmill-protocol are presented in the revised manuscript as requested. The heterogeneous sample of children made us to individualize the test-protocol with larger improvements in inclination compared to speed for some of the heaviest children.
Did you test the reproducibility of the protocols?
Thank you for a highly relevant comment, unfortunately we did not test the reproducibility of the treadmill protocol.
Did the authors realize a familiarization session? a warm-up?
All children started with a five minutes familiarization while walking on the treadmill, now added in the revised manuscript.
Minor comments
L60, P2: “ from children 7-17 years…” the sentence is not clear. Please to rewrite.
The sentence has been modified as suggested.
L83: “cpm” to explain
Cpm, counts per minutes has been added.
L93: The objective criteria for maximal effort included at least 2 of the following: (1) increased workload without corresponding increase in oxygen consumption; (2) respiratory exchange quotient”. What about the plateau of VO2? Do you observed a plateau of VO2 at the end of the treadmill test in your children?
We acknowledge this comment. Criteria defining maximal effort or VO2max in children have been debated for decades and still there is no consensus. Plateau of VO2max has been postulated as one of several criteria to consider when judging whether VO2max has been reached or not. However, in several subjects, and particular children and unfit subjects, a plateau in VO2 are often not confirmed. We have chosen to use the term VO2peak since our test protocol not where designed primarily to establish a plateau in VO2. In addition, we do not have access to raw-data to look into to this further.
L94,P3: How many sessions of 60-min the children realized?
During the five months intervention period with sessions twice a week, the mean attendance in the active play sessions was 60 % with 23 % of the subjects attending less than 45 % of the sessions. We have provided this information in the revised manuscript.
L150,P3: write “where the motivation” instead of “where to”
Thank you, now corrected in the revised manuscript.
Reviewer 3 Report
I had an opportunity review a nice manuscript entitled “Do obese children achieve maximal heart rate during treadmill running?” by Dr. Berntsen et al. The paper describes important aspects of determination of maximal heart rate and exercise testing. The topic is important as heart rate is often used to prescribe exercise and underestimating/overestimating exercise intensity may have effect on optimal exercise intensity, which may have unfavourable consequences on exercise adherence especially among overweight and obese children and adolescents. The Authors found that maximal exercise test on a treadmill elicited a lower maximal heart rate than physically demanding play.
General comments
The Manuscript was generally well written, appropriate references were used, and utilised appropriate statistics to answer the questions.
As the maximal intensity in the treadmill exercise test was one key in the manuscript, I suggest that the Authors re-consider and discuss whether youth really achieved their maximal levels. Maximal effort is usually defined a plateau of VO2 during exercise regardless of increased workload but as few children show a plateau, secondary criteria has been developed. However, Barker et al. (Barker, Williams, Jones, & Armstrong, 2011) among others have questioned VO2peak and traditional secondary criteria in establishment of maximal effort. Please extend your discussion maybe provide more information on effort during exercise test.
Specific comments
Introduction
L42–44. I suggest that the Authors provide some examples on HR thresholds used/needed for example improve cardiorespiratory fitness in children and adolescents. It would provide an informative example how important is to assess HR appropriately as the findings among paediatric studies suggest that children should exercise at least 80% of their max HR to improve their fitness levels.
Methods
L55. I suggest that the Authors use YOUTH instead of children when describing the participants. It would be more accurate as children refers to youth<13 year of age.
L60. Do you have any reference to Norwegian growth charts?
L62. …physicalLY active…
L73. Suggestion: was measured wearing a light underwear to the…
L83. Just a question that does not actually affect the results or interpretation. Is that 2000 cpm validated among obese youth or is it just taken from validation studies performed in lean children? It is well known that accelerometry does not separate exercise intensity between low and high fit children although a certain activity (e.g. stair stepping) is less demanding among high fit children than among low fit children.
L91. Please specify how valid 6–20 RPE scale is in children and explain whether and how it was explained for children before CPET. It is common to anchor perceived exertion to some known or previously perceived exertion. That would be especially important as some overweight and obese children may have not reach vigorous intensity very often so they may not have experiences about maximal effort.
L93. Please see general comment on determination of maximal effort.
L98. Did the Authors used Vantage V or M? Please specify whether wrist-measured or chest-measured heart rate was used.
L101–106. Were there any differences between boys and girls or between children and adolescents? Did you test any interactions?
Results
Table 2. Please consider also reporting VO2peak normalised for DXA-derived lean mass. That would provide physiologically more relevant information (Loftin, Sothern, Abe, & Bonis, 2016; Welsman & Armstrong, 2018) and would benefit future studies to determined LM-scaled fitness levels among obese youth.
Figure 1. please correct as activE play.
Discussion
L161. Questioning the heart rate zones can be done and the Authors have discussed nicely the consequences of underestimated maximal heart rate. 147 bpm would be on average 75% of max HR and would be intense for some youth but not for all. I suggest that the Authors extend the discussion a little bit discussion determining vigorous intensity based on HR at ventilatory or lactate threshold, as it is the gold standard of exercise prescription in exercise physiology.
Literature
Barker, a R., Williams, C. a, Jones, a M., & Armstrong, N. (2011). Establishing maximal oxygen uptake in young people during a ramp cycle test to exhaustion. British Journal of Sports Medicine, 45(6), 498–503. https://doi.org/10.1136/bjsm.2009.063180
Loftin, M., Sothern, M., Abe, T., & Bonis, M. (2016). Expression of VO2peak in Children and Youth, with Special Reference to Allometric Scaling. Sports Medicine (Auckland, N.Z.), (May). https://doi.org/10.1007/s40279-016-0536-7
Welsman, J., & Armstrong, N. (2018). Interpreting Aerobic Fitness in Youth : The Fallacy of Ratio Scaling. Pediatr Exerc Sci, Ahead of print.
Author Response
Reviewer 3
General comments
The Manuscript was generally well written, appropriate references were used, and utilised appropriate statistics to answer the questions.
As the maximal intensity in the treadmill exercise test was one key in the manuscript, I suggest that the Authors re-consider and discuss whether youth really achieved their maximal levels. Maximal effort is usually defined a plateau of VO2 during exercise regardless of increased workload but as few children show a plateau, secondary criteria has been developed. However, Barker et al. (Barker, Williams, Jones, & Armstrong, 2011) among others have questioned VO2peak and traditional secondary criteria in establishment of maximal effort. Please extend your discussion maybe provide more information on effort during exercise test.
We acknowledge this comment and refer to answer to reviewer 2 in addition to in the revised manuscript (discussion) with respect to this.
Specific comments
Introduction
L42–44. I suggest that the Authors provide some examples on HR thresholds used/needed for example improve cardiorespiratory fitness in children and adolescents. It would provide an informative example how important is to assess HR appropriately as the findings among paediatric studies suggest that children should exercise at least 80% of their max HR to improve their fitness levels.
We have provided some more information in the revised manuscript.
Methods
L55. I suggest that the Authors use YOUTH instead of children when describing the participants. It would be more accurate as children refers to youth<13 year of age.
Thank you for a relevant comment. However, we have according to MeSH-terms (NIH, US) consequently used the term child of persons 6-12 yrs of age and adolescents of persons 13-18 yrs of age. In the present manuscript children and adolescents of the included participants 7-17 yrs of age.
L60. Do you have any reference to Norwegian growth charts?
The reference to the Norwegian growth charts is Knudtzon J, Waaler PE, Skjaerven R, Solberg LK, Steen J. New Norwegian percentage charts for height, weight and head circumference for age groups 0–17 years. Tidsskr Nor Laegeforen 1988; 108: 2125–35, now included in the revised manuscript.
L62. …physicalLY active…
Thank you, we have corrected.
L73. Suggestion: was measured wearing a light underwear to the…
Thank you and corrected in the revised manuscript.
L83. Just a question that does not actually affect the results or interpretation. Is that 2000 cpm validated among obese youth or is it just taken from validation studies performed in lean children? It is well known that accelerometry does not separate exercise intensity between low and high fit children although a certain activity (e.g. stair stepping) is less demanding among high fit children than among low fit children.
Physical activity is not an outcome in the present study. We are not aware of any validation studies where cut-point have been established for obese children and adolescents in comparison with normal weight children.
L91. Please specify how valid 6–20 RPE scale is in children and explain whether and how it was explained for children before CPET. It is common to anchor perceived exertion to some known or previously perceived exertion. That would be especially important as some overweight and obese children may have not reach vigorous intensity very often so they may not have experiences about maximal effort.
We acknowledge this comment. The 6-20 RPE scale was briefly explained to the children before the treadmill running. Whether the most unfit or obese children/adolescents understood perceived exertion differently is not known.
L93. Please see general comment on determination of maximal effort.
Please see previous answer (general comment) with respect to this.
L98. Did the Authors used Vantage V or M? Please specify whether wrist-measured or chest-measured heart rate was used.
A relevant question, since we do not have access to these monitors now, I really do not know whether the instructors used a V or a M model. Chest measured heart rate was collected, information added in the revised manuscript.
L101–106. Were there any differences between boys and girls or between children and adolescents? Did you test any interactions?
We are not sure what kind of differences between boys and girls nor interactions the reviewer are asking for. However, due to the limited sample size we did not present stratified analysis for our outcome variable peak HR.
Results
Table 2. Please consider also reporting VO2peak normalised for DXA-derived lean mass. That would provide physiologically more relevant information (Loftin, Sothern, Abe, & Bonis, 2016; Welsman & Armstrong, 2018) and would benefit future studies to determined LM-scaled fitness levels among obese youth.
We acknowledge this comment but since VO2max is not an outcome in the present study and most studies still present VO2max as ml ∙ kg-1 ∙ min-1 we present our data in the same format for comparison.
Figure 1. please correct as activE play.
Thank you, already corrected.
Discussion
L161. Questioning the heart rate zones can be done and the Authors have discussed nicely the consequences of underestimated maximal heart rate. 147 bpm would be on average 75% of max HR and would be intense for some youth but not for all. I suggest that the Authors extend the discussion a little bit discussion determining vigorous intensity based on HR at ventilatory or lactate threshold, as it is the gold standard of exercise prescription in exercise physiology.
Thank you for your comment. HR at ventilatory or lactate threshold are a highly relevant measure of exercise intensity, though it could be questioned if a “gold standard” exists. In a recent published meta-analysis (Braaksma et al. J Sci Med Sport 2018) summarizing physical activity/exercise interventions, also with respect to exercise intensity, in children/adolescents and the effect on cardiorespiratory fitness, maximal aerobic speed, % of HR and % of VO2max were the only measures of intensity reported.
Round 2
Reviewer 3 Report
The Authors have answered all my questions.